# The Gaseous Ozone Therapy as a Promising Antiseptic Adjuvant of Periodontal Treatment: A Randomized Controlled Clinical Trial

**DOI:** 10.3390/ijerph19020985

**Published:** 2022-01-16

**Authors:** Biagio Rapone, Elisabetta Ferrara, Luigi Santacroce, Skender Topi, Antonio Gnoni, Gianna Dipalma, Antonio Mancini, Marina Di Domenico, Gianluca Martino Tartaglia, Antonio Scarano, Francesco Inchingolo

**Affiliations:** 1Interdisciplinary Department of Medicine, “Aldo Moro” University of Bari, 70121 Bari, Italy; giannadipalma@tiscali.it (G.D.); dr.antonio.mancini@gmail.com (A.M.); francesco.inchingolo@uniba.it (F.I.); 2Complex Operative Unit of Odontostomatology, Hospital S.S. Annunziata, 66100 Chieti, Italy; igieneeprevenzione@gmail.com (E.F.); luigi.santacroce@uniba.it (L.S.); 3Department of Clinical Disciplines, School of Technical Medical Sciences, University A. Xhuvani, 3001 Elbasan, Albania; skender.topi@uniel.edu.al; 4Department of Basic Medical Sciences, Neurosciences and Sense Organs, “Aldo Moro” University of Bari, 70121 Bari, Italy; gnoniantonio@gmail.com; 5Department of Precision Medicine, University of Campania Luigi Vanvitelli, 80138 Naples, Italy; marina.didomenico@unicampania.it; 6UOC Maxillo-Facial Surgery and Dentistry, Department of Biomedical, Surgical and Dental Sciences, School of Dentistry, Fondazione IRCCS Ca Granda, Ospedale Maggiore Policlinico, University of Milan, 20100 Milan, Italy; gianluca.tartaglia@unimi.it; 7Department of Oral Science, Nano and Biotechnology and CeSi-Met University of Chieti-Pescara, 66100 Chieti, Italy; ascarano@unich.it

**Keywords:** gaseous ozone therapy, ozone, non-surgical periodontal treatment, moderate periodontitis, severe periodontitis, periodontal disease

## Abstract

Background: the establishment of periodontitis is regulated by the primary etiological factor and several individual conditions including the immune response mechanism of the host and individual genetic factors. It results when the oral homeostasis is interrupted, and biological reactions favor the development and progression of periodontal tissues damage. Different strategies have been explored for reinforcing the therapeutic effect of non-surgical periodontal treatment of periodontal tissue damage. Gaseous ozone therapy has been recognized as a promising antiseptic adjuvant, because of its immunostimulating, antimicrobial, antihypoxic, and biosynthetic effects. Then, we hypothesized that the adjunct of gaseous ozone therapy to standard periodontal treatment may be leveraged to promote the tissue healing response. Methods: to test this hypothesis, we conducted a prospective randomized study comparing non-surgical periodontal treatment plus gaseous ozone therapy to standard therapy. A total of 90 healthy individuals with moderate or severe generalized periodontitis were involved in the study. The trial was conducted from September 2019 to October 2020. Forty-five patients were randomized to receive scaling and root-planning (SRP) used as conventional non-surgical periodontal therapy plus gaseous ozone therapy (GROUP A); forty-five were allocated to standard treatment (GROUP B). The endpoint was defined as the periodontal response rate after the application of the ozone therapy at 3 months and 6 months, defined as no longer meeting the criteria for active periodontitis. Statistical analysis was performed employing SPSS v.18 Chicago: SPSS Inc. Results: periodontal parameters differed significantly between patients treated with the two distinct procedures at 3 months (*p* ≤ 0.005); a statistically significant difference between groups was observed from baseline in the CAL (*p* ≤ 0.0001), PPD (*p* ≤ 0.0001) and BOP (*p* ≤ 0.0001) scores. Conclusions: The present study suggests that SRP combined with ozone therapy in the treatment of periodontitis revealed an improved outcome than SRP alone.

## 1. Introduction

Periodontal disease (PD) is one of the most common inflammatory illnesses affecting the individuals, and the global burden of periodontal disorders, as measured in prevalence, is between 20 and 50%, with severe periodontitis affecting 11.2% worldwide [1,2,3]. The term encompasses a wide spectrum of pathological conditions, ranging from reversible gingival inflammation to severe form, characterized by progressive destruction of alveolar bone [3]. All clinical manifestations have the same pathogenic pathway, with a dramatic increase in bacterial pathogens aggregation (bacterial plaque) as a mainly etiologic factor and important genetic and immunoregulatory individual determinants of the severity of the disease [4,5,6,7,8,9]. In general, the conventional treatment for periodontal lesions is a mechanical and manual non-surgical procedure, named scaling and root planning (SRP), aimed at eliminating supra and sub-gingival bacterial plaque and calculus [10,11]. Several studies have examined the application of add-on therapy in the treatment of periodontitis (e.g., laser or photodynamic therapies) to improve immunogenic responses [12,13,14,15,16,17], and inter-individual variability of response to various adjuvant treatments and therapeutic procedures has been widely reported [18,19,20,21]. Recently, the treatment with gaseous ozone has been studied as a support for SRP for its important effects of immune modulation and healing [22,23,24]. Ozone therapy has been extensively studied in medicine because of its physicochemical properties and its unbelievable versatility for many biomedical applications, specifically degenerative, neurological, orthopaedic and genitourinary disorders [25,26,27,28,29,30]. Also in dentistry, it has an extensive application which fluctuates from endodontia to conservative as well as the treatment of tooth sensitivity [31,32,33,34]. In the context of periodontal infection, oxygen/ozone gas can act as a powerful device for the targeted antiseptic action, potentially reducing the impact of microbial burden, and contemporary increasing the immune system capability [18,27,28,29,30,31,32,33,34,35]. The efficacy of ozone (O_3_) is greatly related to the beneficial chemical and physical properties, that makes it eligible for employment in periodontal area [12,19,30]. As extensively documented, the ozone has an immunomodulatory, anti-inflammatory and biocide action [36,37,38]. Its antiseptic activity is mediated by disruption of bacterial cell membrane integrity, resulting in their lysis and death [39,40]. In addition, the ozone exerts a double damage: on sulfhydryl groups of specific enzymes, disrupting the normal cellular enzymatic activity and diminishing their function; on the base components of nucleic acids, the purines and pyrimidine, resulting in damage to DNA [41,42]. The anti-inflammatory property is due to the disruption of the self-perpetuating inflammatory cycle altering the breakdown of Arachidonic acid-derived prostaglandins that contribute to the development of inflammation [43,44,45]. Furthermore, O_3_ contributes to activate the immune cells and it is involved in the production of cytokines [19,25,40]. According to these pharmacological properties, the purpose of this trial was to determine the effectiveness of gaseous ozone therapy in patients with periodontitis, by assuming the superiority of treatment respect to the SRP only. To test this hypothesis, we conducted a prospective randomized controlled study investigating the effectiveness of gaseous ozone therapy in patients with moderate and severe periodontitis.

## 2. Materials and Methods

### 2.1. Ethical Considerations

The protocol was conducted in compliance with the Ethics Committee Approval INTL_ALITMKCOOP/HealthMicroPath/HMM2019_IPM and according to Good Clinical Practice and the Declaration of Helsinki Declaration of 1975, as revised in 2013 [46]. Written informed consent was obtained from all patients before the study.

### 2.2. Study Design and Participants

The trial was conducted from September 2020 to October 2021 at School of Technical Medical Sciences, University A. Xhuvani, Elbasan, Albania. This randomized double-masked clinical trial was carried out to test the hypothesis that the gaseous ozone therapy application as adjunct to SRP leads to significant improvements of periodontal parameters compared with SRP alone. Periodontal examinations were performed by one blinded examiner (BR), and three operators (dentist, AS; dentist, FI, dental hygienist, EF) carried out the treatment at each time point. One blinded statistician (AG) performed the data analysis. Based on limited data available at the time, a sample size of 80 participants (40 for each group) were required to achieve 90% study power using a two-group t-test assuming an α-level of 0.05. Considering a drop-out rate of 10% total sample size, was planned a recruitment of 90 participants.

### 2.3. Inclusion and Exclusion Criteria

Adult patients were considered eligible for inclusion in the trial if they had a diagnosis for moderate-to-severe periodontitis. Periodontitis diagnosis was determined according to the new criteria presented in the World Workshop on the Classification of Periodontal and Peri-implant Diseases and Conditions [2]. Qualifying patients met all the following inclusion criteria: having a periodontitis diagnosis; having at least 16 teeth with a minimum of four teeth in each quadrant; men and women, aged ≥18 years; could provide informed consent. Reasons for non-enrolment were the following: unable to meet the inclusion criteria (1) underwent administration of any systemic antibiotic regimen within the previous 6 months before enrolment (2); having undergone periodontal therapy within the 12 months prior to the randomization (3); history of systemic diseases (4); medical conditions that contraindicated ozone therapy (e.g., respiratory diseases) (5); current daily smokers with a number of >10 cigarettes/day (6); cognitive or serious mental illness; and pregnancy (7). The source population for this study consisted of subjects with a mean age of 51.56 ± 10.35. The follow-up started on the date of the baseline and ended at 6 months; an interim analysis report addressing the impact of ozone therapy on periodontal outcomes was conducted at 14 days.

### 2.4. Randomization and Blinding

Eligible patients were randomly assigned, in a 1:1 ratio, to following groups: SRP + OZONE (**Test Group A**, *n* = 45); SRP (**Control Group B**, *n* = 45). Examiners and statistician were blinded to group assignment. Randomization was performed with computer generated random number list. At baseline, at 3 and 6 months after SRP each patient received periodontal examination by two calibrated examiners, blinded to the treatment group. The examiners calibration was conducted before the study. The alignment exercise resulted in 80% inter-examiner reliability and 90% intra-examiner reproducibility [41].

### 2.5. Periodontal Clinical Parameters Measurement

To assess the periodontal status before and subsequent the intervention and infer the difference inter-groups, the following clinical outcome parameters were revealed: Bleeding on probing (BOP) was recorded to assess gingival inflammation and it was registered as the percentage at four sites per tooth showing bleeding 30 s after probing [47]; Probing pocket depth (PPD), which is established by calculating the distance from the gingival margin (GM) to the base of the sulcus/pocket with a calibrated periodontal probe; and clinical attachment level (CAL), which is determined by measuring the distance from the cemento-enamel-junction (CEJ) to the base of the sulcus/pocket [41]. Probing pocket depth and CAL were recorded at six sites per tooth. The assessment of clinical status was carried out employing the standard probing measurements using a marked periodontal probe (UNC15 probe, Hu Friedy, Chicago, IL, USA).

### 2.6. Outcomes

The outcomes were the probing pocket depth reduction, clinical attachment level improvement at 3 and 6 months.

### 2.7. Treatment

After enrolment, for each patient of both groups, scaling and root planning (SRP) treatment was performed. The objective of scaling is to remove supra- and sub-gingival calculus deposits and root planning to smooth root surfaces.

Each participant received hygiene education session and appropriate motivation and underwent supra- and subgingival prophylaxis with ultrasonic instruments. All sites with probing pocket depth (PPD) ≥4 mm were root planed by using manual instruments (Gracey curets, Hu Friedy, Chicago, IL, USA), under local anaesthesia. No rinsing with chlorhexidine digluconate solution was recommended so as not to affect the results. Follow up was planned in three- and six-months’ time. In the group assigned to SRP + OZONE, gaseous ozone treatment was performed in three steps after instrumentation by ultrasonic instruments employing an ozone generator (Ozone DTA, **Sweden** & **Martina Company**; Carrara San Giorgio, Veneto, Italy), according to manufacturer instructions, as follows: **Step 1**. 2-min rinse with ozonated water at a ratio of 1:3; Full-mouth decontamination; Topical irrigation with ozonated water; 1–2 cycles of ozone gas at 8–10 power in correspondence of pathological pockets, under local anaesthesia. **Step 2**. Quadrant root planning; 2-min rinse with ozonated water at a ratio of 1:3; Deplaquing; 1–2 cycles of ozone gas at 8–10 power in correspondence of pathological pockets for each quadrant, under local anaesthesia. **Step 3**. Maintenance: 2-min rinse with ozonated water at a ratio of 1:3; Deplaquing; 1–2 cycles of ozone gas at 4–5 power in correspondence of pathological pockets for all quadrants, two weeks after completion of treatment.

To maintain a state of optimal periodontal health for a correct view of the periodontal ligament, each patient was motivated to a correct home management using a roto oscillating or sonic toothbrush and toothpastes to keep the periodontium intact and avoid the progression of the disease with the destruction of the tissue itself [48,49].

### 2.8. Statistical Analysis

The Shapiro-Wilk test was used to confirm normal distribution of the data related to each numerical variable for each follow-up time point. Continuous variables were presented as mean ± standard deviation (SD) and all categorical data are expressed as a frequency or a percentage. The comparison of data from the two groups at each time point was performed by using the unpaired 2-sample *t* test. A mixed model multivariate analysis of covariance (MANCOVA) with two within-subjects factors and one between-subjects factor was conducted to determine whether significant differences exist among the time points for PPD and CAL between the levels of treatment (SRP+ OZONE or SRP) after controlling for stage of disease (moderate or severe), age, and sex as covariates. A *p* value of <0.05 was considered statistically significant. Statistical analysis was performed employing SPSS Statistics for Windows, version 18 (SPSS Inc., Chicago, IL, USA).

## 3. Results

The enrollment was started in September 2019 and ended in December 2019. During this phase, 232 patients were screened. Figure 1 shows the trial profile.

A total of 90 patients were included in the study. Baseline demographic characteristics and clinical periodontal parameters of the 90 patients included in our analysis are illustrated in Table 1 and Table 2, respectively.

Mean age was 51.62 ± 14.42 for Group A and 49.88 ± 10.54 for the Control Group. Twenty-two percentage of the patients had moderate periodontitis, 78% had diagnosis of severe periodontitis. As shown in Table 2, no significant difference was detected between the two groups in mean score of two periodontal parameters at baseline (PPD 5.39 vs. 5.37, *p* = 0.81; BOP 49 vs. 50.83, *p* = 0.62). However, the Control group showed a higher mean CAL score than the Test group (5.78 vs. 5.53, *p* ≤ 0.0002). At 3 months a statistically significant difference in the PPD (*p* ≤ 0.0001), CAL (*p* ≤ 0.003) and BOP (*p* ≤ 0.0001) was observed between the groups, as shown in Table 3.

At 6 months a significant decrease was observed in the PPD, CAL and BOP (*p* ≤ 0.0001, *p* ≤ 0.0001, and *p* ≤ 0.0001 respectively) in the test group compared to control group (Table 4).

Data resulting from the Unpaired *T* test are shown in Table 5 and Figure 2.

### 3.1. MANCOVA Analysis

#### 3.1.1. Assumptions

**Normality.** The assumption of normality was assessed by plotting the quantiles of the model residuals against the quantiles of a Chi-square distribution. Figure 3 presents a Q-Q scatterplot of model residuals.

**Homoscedasticity.** Homoscedasticity was evaluated by plotting the residuals against the predicted values [48,49,50]. Figure 4 presents a scatterplot of predicted values and model residuals.

**Sphericity.** Mauchly’s test was used to assess the assumption of sphericity [49]. The results showed that the variances of difference scores across the levels of Time Factor were all similar based on an alpha of 0.05, *p* = 0.483, indicating the sphericity assumption was met for Time Factor. The results showed that the variances of difference scores across the levels of Time Factor:Dipendent variable (Dv) Factor were significantly different from one another based on an alpha of 0.05, *p* < 0.001, indicating the sphericity assumption was violated for Time Factor:Dv Factor.

**Multivariate Outliers.** To identify influential points in the residuals, Mahalanobis distances were calculated and compared to a χ^2^ distribution [51]. An outlier was defined as any Mahalanobis distance that exceeds 22.46, the 0.999 quantile of a χ^2^ distribution with 6 degrees of freedom [52]. There were no outliers detected in the model.

**Homogeneity of regression slopes.** The assumption for homogeneity of regression slopes was assessed by rerunning the mixed model MANCOVA, but this time including interaction terms between each independent variable and covariate [49,50]. The model with covariate-independent variable interactions did not explain significantly more variance in the dependent variables than the original model, *F*(18, 207) = 1.1, *p* = 0.356. This implies that none of the covariates interacted with the independent variables and the assumption of homogeneity of regression slopes was met.

**Covariate-IV independence.** An ANOVA was conducted for each pair of numeric covariates and independent variables to assess independence [49]. A multinomial regression model was conducted and compared to the null model for each pair of categorical covariates and independent variables to assess independence. There were no significant models for any combination of covariates and independent variables based on an alpha of 0.05, indicating the assumption of independence between covariates and independent variables was met.

#### 3.1.2. Mixed Model MANCOVA Results

The results were examined based on an alpha of 0.05. Table 6 presents the MANCOVA results.

The *p*-values for and any interaction with these within-subjects factors were calculated using the Greenhouse-Geisser corrections to adjust for the violation of the sphericity assumption.

**Between-Subjects.** The main effect for Treatment was significant *F*(1, 75) = 23.28, *p* < 0.001, indicating that there were significant differences in PPD and CAL between the levels of Treatment after controlling for stage of disease, age, and sex. The covariate, Stage_of_disease, was significantly related to PPD and CAL, *F*(1, 75) = 4.73, *p* = 0.033. The covariate, age, was significantly related to PPD and CAL, *F*(1, 75) = 7.69, *p* = 0.007. The covariate, sex, was not significantly related to PPD and CAL, *F*(1, 75) = 2.43, *p* = 0.123.

**Within-Subjects.** The main effect for Time Factor was significant *F*(2, 150) = 8.11, *p* < 0.001, indicating there were significant differences in PPD and CAL across the levels of Time Factor ignoring Dv Factor after controlling for stage of disease, age, and sex. The main effect for Dv Factor was not significant *F*(1, 75) = 0.86, *p* = 0.358, indicating the values for across the levels of Dv Factor, PPD and CAL, were all similar regardless of Time Factor after controlling for stage of disease, age, and sex. The main effect for Time Factor and Dv Factor was not significant *F*(2, 150) = 0.20, *p* = 0.778, indicating that the relationships between the levels of Dv Factor were similar across the levels of Time Factor after controlling for Stage of disease, age, and sex.

**Within-Between Interactions.** The interaction effect between Time Factor and treatment was significant *F*(2, 150) = 25.45, *p* < 0.001, indicating that the relationships between the levels of Time Factor differed significantly between the levels of treatment ignoring Dv Factor after controlling for stage of disease, age, and sex.

The interaction effect between Dv Factor and Treatment was significant *F*(1, 75) = 8.74, *p* = 0.004, indicating that the relationships between the levels of Dv Factor differed significantly between the levels of treatment regardless of Time Factor after controlling for stage of disease, Age, and sex.

The interaction effect between Time Factor, Dv Factor, and treatment was significant *F*(2, 150) = 9.57, *p* < 0.001, indicating that the relationships between the combinations of Time Factor and Dv Factor differed significantly between the levels of treatment after controlling for stage of disease, age, and sex.

**Within-Covariate Interactions.** The interaction effect between Time Factor and stage of disease was not significant, *F*(2, 150) = 1.82, *p* = 0.166, indicating that the relationships between the levels of Time Factor were similar for all values of stage of disease. The interaction effect between Time Factor and age was not significant, *F*(2, 150) = 0.73, *p* = 0.482, indicating that the relationships between the levels of Time Factor were similar for all values of age. The interaction effect between Time Factor and sex was not significant, *F*(2, 150) = 1.32, *p* = 0.271, indicating that the relationships between the levels of Time Factor were similar between the levels of sex.

The interaction effect between Dv Factor and stage of disease was not significant, *F*(1, 75) = 0.26, *p* = 0.611, indicating that the relationships between the levels of Dv Factor were similar for all values of stage of disease. The interaction effect between Dv Factor and age was not significant, *F*(1, 75) = 0.15, *p* = 0.698, indicating that the relationships between the levels of Dv Factor were similar for all values of age. The interaction effect between Dv Factor and sex was not significant, *F*(1, 75) = 0.47, *p* = 0.494, indicating that the relationships between the levels of Dv Factor were similar between the levels of sex.

The interaction effect between Time Factor, Dv Factor, and Stage of disease was not significant, *F*(2, 150) = 0.03, *p* = 0.955, indicating that the relationships between the combinations of Time Factor and Dv Factor were similar for all values of stage of disease. The interaction effect between Time Factor, Dv Factor, and age was not significant, *F*(2, 150) = 0.42, *p* = 0.617, indicating that the relationships between the combinations of Time Factor and Dv Factor were similar for all values of age. The interaction effect between Time Factor, Dv Factor, and sex was not significant, *F*(2, 150) = 1.29, *p* = 0.276, indicating that the relationships between the combinations of Time Factor and Dv Factor were similar between the levels of sex.

## 4. Discussion

The key to onset and progression of periodontitis consists of two canonical pathways: the oral microbial subversion, the central stimulus, resulting in the expression of proinflammatory cytokines to eradicate pathogens and repair the damage tissues; in parallel, the genetic, environmental and systemic health status which contribute cumulatively to the disease etiology and development. The goal of periodontal therapy is based on the eradication of pathogenic bacteria responsible for the onset of the disease to control the inflammatory. The aim of this clinical trial was to determine the impact of gaseous ozone therapy in conjunction to conventional periodontal treatment on conditions and severity of periodontal disease in healthy subjects diagnosed with moderate or severe periodontitis, in comparison with standard treatment. To provide a compelling comparison be-tween the two therapies, a randomized controlled trial was designed. Ozone therapy is a practice of complementary medicine and its effects have been widely confirmed [14,48,49,50]. Beginning in the 1960 [51], multiple trials assessed the safety and efficacy of ozone in medicine for several therapeutic indications. Humans’ studies have exposed the biological plausibility of ozone-induced beneficial impact on several pathological conditions [43,44,45,46,47,48,49] and described the mechanism of action of ozone, which encompasses the capacity to inactivate bacteria, viruses, fungi, yeast and protozoa by disrupting the integrity of the bacterial cell; the ability to stimulate the increase in the red blood cell glycolysis rate; the capacity to activate the immune response by causing the increase in the production of interleukin-2 which determines a cascade of subsequent immunological reactions [17,52,53,54]. Application methods include in-direct and direct procedures, such as the intramuscular injection, ozone bag and others. In dentistry, the indirect technical methods including the ozonated water, ozonated oil and gaseous ozone generator are employed. The aqueous (1.25–20 µmgL^−1^) and gaseous ozone (1–53 g m^−3^) are predominantly employed against periodontopathogenic and endodontic bacteria, including the Enterococcus faecalis, the mainly endodontic pathogen [55,56]. Boch et al., reported 85.38% reduction of bacterial count after gaseous ozone application on *Enterococcus faecalis* biofilm in root canals and 99.5% eradication of bacteria when the ozone was combined with NaOCl [56]. Case et al., demonstrated the efficacy of ozone combined with ultrasonic agitation and ozone alone on *E. fecalis* [57]. Further, the antimicrobial activity of the ozone has been documented against the Staphylococcus aureus and Staphylococcus epidermidis [13], registering a significantly de-crease in absolute counts of microorganisms. Emerging studies have examined gaseous ozone therapy in addition to non-surgical periodontal therapy. The rationale behind the use of ozone therapy is based on the concept of the specifically inflammatory target pathway, as well as the antimicrobial activity. The microbial pathogenesis of periodontitis and the immune response are the two determinants of this choice. Our results support previous recent studies showing that patients who have been treated with ozone exhibited statistically and clinically significant improvement in periodontal inflammation after gaseous ozone treatment [22,28]. In our previous study of diabetic patients with periodontitis, we observed a sensitive improvement of periodontal status after the application of gaseous ozone [28]. The significant difference between the two groups in a decrease in periodontal outcomes at 3 months in the gaseous ozone-treated group rationalize the improvement of the periodontal stability condition at 6 months in test groups. We hypothesized that the significant reduction displayed in the test group may reflect the biological activity on periodontal tissues and hence an improvement on disease [15,16,58,59]. We theorized that the antimicrobial activity is a key step. Periodontitis is a disease whose course essentially feeds on the presence of pathogenic bacteria that alter homeostasis and induce the establishment of the disease. Then, we assumed that the ozone therapy might promote the healing consequent improvement of the state of the disease by stimulating the immune response and a more rapid lowering of the microbial load. Our results are in contrast with findings reported by Tasdemir et al. [60], which are also based on topical gaseous ozone application into periodontal pockets. They reported no significant differences between the two groups during the follow up in periodontal parameters. Although significant differences in CAL between the groups, some factors in our study could be considered, plausible interindividual differences and the time between ozone treatment and the subsequent 3 months could reveal a different healing pattern in these individuals. The addition of ozone treatment showed a marked improvement in periodontal conditions compared with the test group, while both groups manifested a significant reduction in pocket probing at 3 months. Each patient received the same oral hygiene instruction and motivation, and this may influence long-term outcomes. The current study has some limitations, first, the use of ozone therapy was only granted during the SRP phase, without any recall, and the follow-up was limited to explore the potential benefit of recall. The limitation of this study could be related to the use of criteria to define success as changes in PD and CAL [61,62,63,64,65,66,67], because of the potential limited representativeness of the effectiveness of ozone therapy, which includes additional benefits. Further studies investigating biochemical parameters of oxidative stress might be useful for a more in-depth evaluation on periodontal tissue healing restoration.

## 5. Conclusions

This randomized clinical trial suggests that gaseous ozone therapy in conjunction with the conventional periodontal treatment may reduce the likelihood of periodontitis advancing. Based on previous research we hypothesized that gaseous ozone treatment of periodontitis, as adjuvant of SRP, may have encouraging therapeutic effects.

## Figures and Tables

**Figure 1 ijerph-19-00985-f001:**
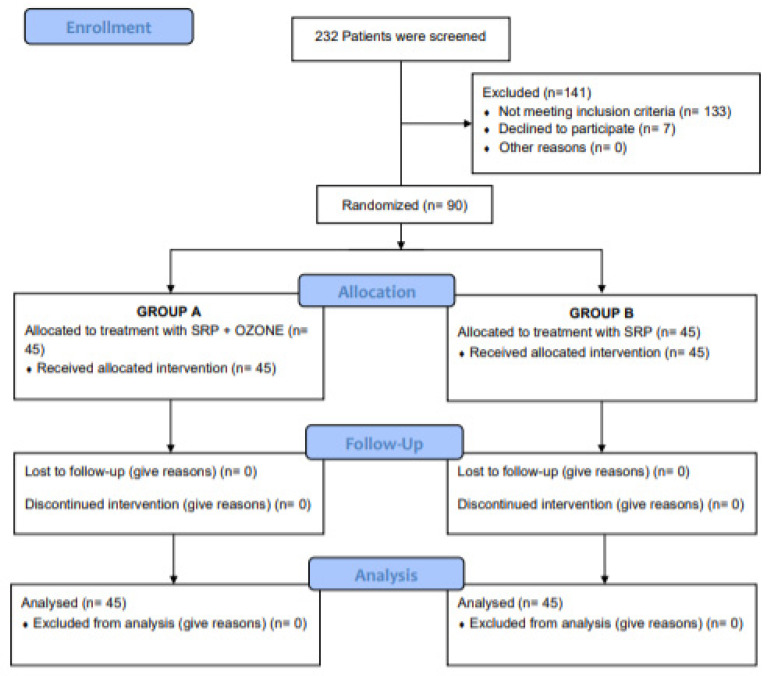
Consort diagram showing the screening, enrolment and randomization of study patients.

**Figure 2 ijerph-19-00985-f002:**
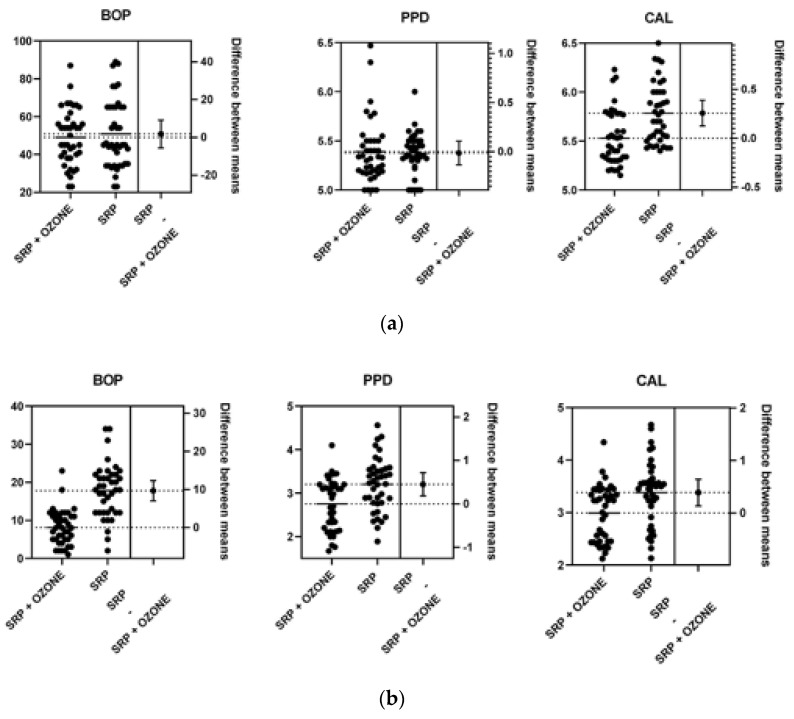
The difference between means at baseline (**a**), 3 (**b**) and 6 (**c**) months.

**Figure 3 ijerph-19-00985-f003:**
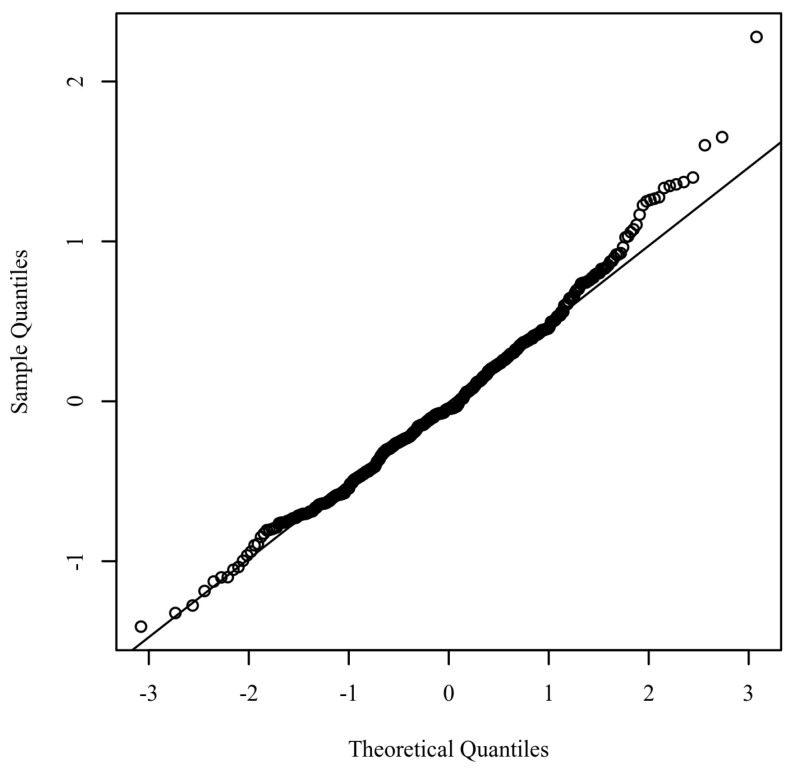
Q-Q scatterplot for normality of the residuals for the regression model.

**Figure 4 ijerph-19-00985-f004:**
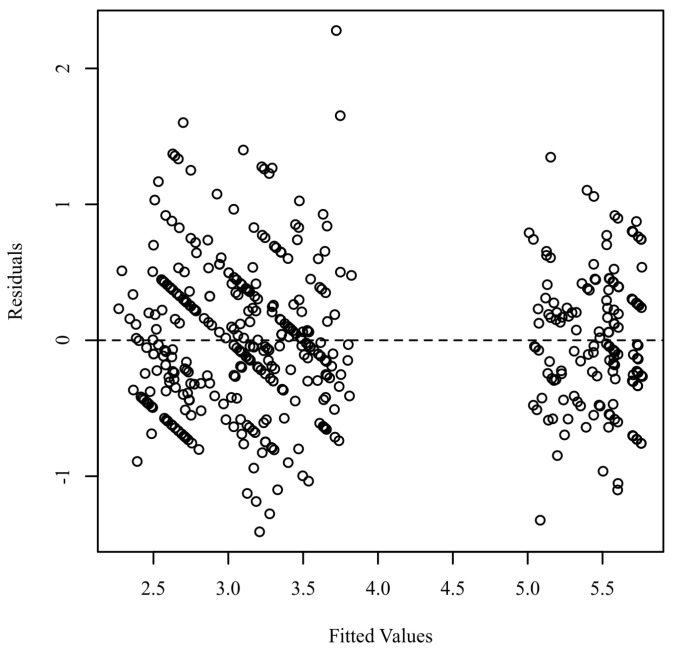
Residuals scatterplot testing homoscedasticity.

**Table 1 ijerph-19-00985-t001:** Baseline demographic characteristics of participants.

	Group A *	Group B **
Age (mean ± SD)	51.62 ± 9.56	49.88 ± 10.54
Sex	M 87% F 13%	M 78% F 22%
Prevalence of Moderate Periodontitis (%)	78	83
Prevalence of Severe Periodontitis (%)	22	17

Group A *: Test Group (SRP + OZONE); Group B **: Control Group (SRP).

**Table 2 ijerph-19-00985-t002:** Baseline clinical periodontal parameters of both groups.

	PPD (mm) Group A *	PPD (mm) Group B **	*p* Value	CAL (mm) Group A	CAL (mm) Group B	*p* Value	BOP (%) Group A	BOP (%) Group B	*p* Value
Mean	5.39	5.37	0.81	5.53	5.78	<0.05	49	50.83	0.62
Std. Deviation	0.31	0.2	-	0.27	0.3	-	14.74	18.11	-

Group A *: Test Group (SRP + OZONE); Group B **: Control Group (SRP); PPD: Probing pocket depth; CAL: Clinical attachment level; BOP: Bleeding on probing. *p* Value: statistically significant at <0.05.

**Table 3 ijerph-19-00985-t003:** Change of periodontal clinical parameters of both groups at 3 months.

	PPD (mm) Group A *	PPD (mm) Group B **	*p* Value	CAL (mm) Group A	CAL (mm) Group B	*p* Value	BOP (%) Group A	BOP (%) Group B	*p* Value
Mean	2.75	3.2	< 0.001	2.99	3.38	< 0.003	8.12	17.78	< 0.0001
Median	2.93	3.25	-	3.18	3.47	-	8	18	-
Std. Deviation	0.59	0.6	-	0.53	0.6	-	4.6	7.05	-

Group A *: Test Group (SRP + OZONE); Group B **: Control Group (SRP); PPD: Probing pocket depth; CAL: Clinical attachment level; BOP: Bleeding on probing. *p* Value: statistically significant at <0.05.

**Table 4 ijerph-19-00985-t004:** Change of periodontal clinical parameters of both groups at 6 months.

	PPD (mm) Group A *	PPD (mm) Group B **	*p* Value	CAL (mm) Group A	CAL (mm) Group B	*p* Value	BOP (%) Group A	BOP (%) Group B	*p* Value
Mean	2.67	3.28	<0.0001	2.85	3.42	<0.0001	6.27	12.83	<0.0001
Median	2.52	3.41	-	2.94	3.37	-	6	12	-
Std. Deviation	0.48	0.71	-	0.48	0.75	-	3.32	5.7	-

Group A *: Test Group (SRP + OZONE); Group B **: Control Group (SRP); PPD: Probing pocket depth; CAL: Clinical attachment level; BOP: Bleeding on probing. *p* Value: statistically significant at <0.05.

**Table 5 ijerph-19-00985-t005:** Unpaired *T* test results at 3 and 6 months.

	PPD (mm) 3 Months	PPD (mm) 6 Months	CAL (mm) 3 Months	CAL (mm) 6 Months	BOP (%) 3 Months	BOP (%) 6 Months
*p* value	<0.05	<0.05	<0.05	<0.05	<0.05	<0.05
T value	3.35	4.43	3.06	4.02	7.23	6.27
Df	78	78	78	78	78	78
Differences between the means ± SEM	0.45 ± 0.13	0.6 ± 0.13	0.39 ± 0.12	0.57 ± 0.14	9.65 ± 1.33	6.55 ± 1.04
CI 95%	0.18 to 0.71	0.33 to 0.88	0.13 to 0.64	0.28 to 0.85	6.99 to 12.3	4.47 to 8.62
R	0.12	0.2	0,1	0.17	0.4	0.33
F	1.058	2.14	1.3	2.36	2.34	2.95
*p* value	0.86	0.01	0.4	0.008	0.009	0.001

*p* Value: statistically significant at <0.05.

**Table 6 ijerph-19-00985-t006:** Mixed Model MANCOVA Results.

Source	*df*	SS	MS	F	*p*	*η^2^_p_*
Between-Subjects						
Treatment	1	8.76	8.76	23.28	<0.001	0.24
Stage_of_disease	1	1.78	1.78	4.73	0.033	0.06
Age	1	2.90	2.90	7.69	0.007	0.09
sex	1	0.92	0.92	2.43	0.123	0.03
Residuals	75	28.24	0.38			
Within-Subjects						
Time Factor	2	4.41	2.20	8.11	<0.001	0.10
Treatment:Time Factor	2	13.82	6.91	25.45	<0.001	0.25
Stage_of_disease:Time Factor	2	0.99	0.49	1.82	0.166	0.02
Age:Time Factor	2	0.40	0.20	0.73	0.482	0.01
sex:Time Factor	2	0.71	0.36	1.32	0.271	0.02
Time Factor Residuals	150	40.74	0.27			
Dv Factor	1	0.23	0.23	0.86	0.358	0.01
Treatment:Dv Factor	1	2.38	2.38	8.74	0.004	0.10
Stage_of_disease:Dv Factor	1	0.07	0.07	0.26	0.611	0.00
Age:Dv Factor	1	0.04	0.04	0.15	0.698	0.00
sex:Dv Factor	1	0.13	0.13	0.47	0.494	0.01
Dv Factor Residuals	75	20.43	0.27			
Time Factor:Dv Factor	2	0.11	0.06	0.20	0.778	0.00
Treatment:Time Factor:Dv Factor	2	5.43	2.71	9.57	<0.001	0.11
Stage_of_disease:Time Factor:Dv Factor	2	0.01	0.01	0.03	0.955	0.00
Age:Time Factor:Dv Factor	2	0.24	0.12	0.42	0.617	0.01
sex:Time Factor:Dv Factor	2	0.73	0.37	1.29	0.276	0.02
Time Factor:Dv Factor Residuals	150	42.56	0.28			

**Degrees of Freedom (*df*):** Refers to the number of values used to compute a statistic; an *F*-test has two values for *df*: the first is determined by the number of groups being compared—1, and the second is approximately the number of observations in the sample; used with the *F* to determine the *p*-value; ***F* Ratio (*F*):** The ratio of explained variance to error variance; used with the two *df* values to determine the *p*-value; **Partial Eta Squared (*η^2^_p_*):** Effect size for the ANOVA/MANOVA and determines the strength of the differences among the groups; ***p*-value:** The probability of obtaining the observed results if the null hypothesis is true; **Residuals:** Refers to the difference between the predicted value for the dependent variable and the actual value of the dependent variable.

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
