# Peer review of "The Gaseous Ozone Therapy as a Promising Antiseptic Adjuvant of Periodontal Treatment: A Randomized Controlled Clinical Trial"

_ijerph, 2022, doi:10.3390/ijerph19020985_

Round 1
Reviewer 1 Report
The authors studied gaseous ozone therapy as an alternative antiseptic adjuvant of periodontal treatment.
Periodontal diseases effect most of the individuals in the population. If treated properly the disease can be stopped. I believe this kind of alternative treatments must be understood deeply by clinicians to increase the effect of non-surgical, first step treatments. I think this is a well written manuscript. I suggest it to be accepted as a publication.
Author Response
#Reviever 1
- The authors studied gaseous ozone therapy as an alternative antiseptic adjuvant of periodontal treatment. Periodontal diseases effect most of the individuals in the population. If treated properly the disease can be stopped. I believe this kind of alternative treatments must be understood deeply by clinicians to increase the effect of non-surgical, first step treatments. I think this is a well written manuscript. I suggest it to be accepted as a publication.
Thank you so much for your considerations.
Reviewer 2 Report
In the publication “The gaseous ozone therapy as a promising antiseptic adjuvant of periodontal treatment: a clinical randomized controlled trial” by Biagio Rapone, Elisabetta Ferrara, Luigi Santacroce, Skender Topi, Antonio Gnoni, Gianna Dipalma, Antonio Mancini, Marina Di Domenico, Gianluca Martino Tartagli , Antonio Scarano, Francesco Inchingolo, prepared for being published in Section Oral Health, Collection Frontiers in Oral Health and Health Promotion Research, the authors tried to investigate the Gaseous ozone therapy as a promising antiseptic adjuvant, because of its immunostimulating, antimicrobial, antihypoxic, and bio-synthetic effects. Then, they hypothesized that the adjunct of gaseous ozone therapy to standard periodontal treatment may be leveraged to promote the tissue healing response. As scientific background the authors tried to explain that the establishment of periodontitis is regulated by the primary etiological factor and several individual conditions including the immune response mechanism of the host and individual genetic factors. It results when the oral homeostasis is interrupted, and biological reactions favour the development and progression of periodontal tissues damage. Different strategies have been explored for reinforcing the therapeutic effect of non-surgical periodontal treatment of periodontal tissue damage. The reviewer is missing in the publication an intensive literature discussion due to the topic “Different strategies have been explored for reinforcing the therapeutic effect of non-surgical periodontal treatment of periodontal tissue damage”. a short pubmed research would be very helpful here. So, any reference to recently published papers on the topic is missing:
- Evaluating clinical and laboratory effects of ozone in non-surgical periodontal treatment: a randomized controlled trial. Seydanur Dengizek E, Serkan D, Abubekir E, Aysun Bay K, Onder O, Arife C.J Appl Oral Sci. 2019 Jan 14;27: e20180108. doi: 10.1590/1678-7757-2018-0108.
- The effects of ozone therapy as an adjunct to the surgical treatment of peri-implantitis. Isler SC, Unsal B, Soysal F, Ozcan G, Peker E, Karaca IR. J Periodontal Implant Sci. 2018 Jun 30;48(3):136-151. doi: 10.5051/jpis.2018.48.3.136. eCollection 2018 Jun. PMID: 29984044
To test this hypothesis, the authors conducted a prospective randomized study comparing non-surgical perio-dontal treatment plus gaseous ozone therapy to standard therapy. A total of 90 healthy individuals with moderate or severe generalized periodontitis were involved in the study. The trial was conducted from September 2019 to October 2020. Forty-five patients were randomized to receive scaling and root-planning (SRP) used as conventional non-surgical periodontal therapy plus gaseous ozone therapy (GROUP A); forty-five were allocated to standard treatment (GROUP B). The endpoint was defined as the periodontal response rate after the application of the ozone therapy at 3 months and 6 months, defined as no longer meeting the criteria for active periodontitis. Statistical analysis was performed employing SPSS v.18 Chicago: SPSS Inc. Comparing with the literature the reviewer is missing to evaluate both, clinical and biochemical in vivo effects of gaseous ozone on oxidative stress (TAS and TOS) and TGF-β, a marker for periodontal recovery.
As results the authors revealed that periodontal parameters differed significantly between patients treated with the two distinct procedures at 3 months (P= <0.005); a statistically significant difference between groups was observed from base-line in the CAL (P= <0.0001), PPD (P= <0.0001) and BOP (P= <0.0001) scores. The reviewer agrees the use of ozone in periodontal therapy is based on its antimicrobial, immunostimulating, anti-hypoxic, and biosynthetic properties. Ozone is theorized to contribute to periodontal healing by eliminating pathogens, activating the immune system, and stimulating the humoral antioxidant system, however, clinical evidence thereto is limited. The reviewer is missing a clear outline of the clinical trials strategy in this paper. The reviewer is convinced that the study has been done using a randomized, double-blinded study design.
In conclusions the present study suggests that SRP combined with ozone therapy in the treatment of periodontitis revealed an improved outcome than SRP alone. Ozone has several positive effects on cellular and humoral immune system components, in stimulation of the proliferation of immunocompetent cells and synthesis of immunoglobulins. It has been well-established that ozone activates macrophages and enhances the sensitivity of microorganisms to phagocytosis. Immune cells of the body produce certain cytokines as a response to the activation caused by ozone. Ozone triggers the synthesis of various biologically active substances that reduce inflammation and promote wound healing, such as interleukins, leukotrienes, and prostaglandins. Oxidant and AO molecules are also known to play important roles in the immune response to PD. For this purpose, the authors should evaluate the effects of ozone treatment on oxidants and AOs by assaying levels of TOS, NO, 8-OHdG, MDA, GSH, and TAS in saliva to investigate the ozone effects on the immune response in PD. These newly investigated findings may suggest that the stimulatory effects of ozone administration on periodontal tissues may be limited.
Author Response
#Reviever 2
- In the publication “The gaseous ozone therapy as a promising antiseptic adjuvant of periodontal treatment: a clinical randomized controlled trial” by Biagio Rapone, Elisabetta Ferrara, Luigi Santacroce, Skender Topi, Antonio Gnoni, Gianna Dipalma, Antonio Mancini, Marina Di Domenico, Gianluca Martino Tartagli , Antonio Scarano, Francesco Inchingolo, prepared for being published in Section Oral Health, Collection Frontiers in Oral Health and Health Promotion Research, the authors tried to investigate the Gaseous ozone therapy as a promising antiseptic adjuvant, because of its immunostimulating, antimicrobial, antihypoxic, and bio-synthetic effects. Then, they hypothesized that the adjunct of gaseous ozone therapy to standard periodontal treatment may be leveraged to promote the tissue healing response. As scientific background the authors tried to explain that the establishment of periodontitis is regulated by the primary etiological factor and several individual conditions including the immune response mechanism of the host and individual genetic factors. It results when the oral homeostasis is interrupted, and biological reactions favour the development and progression of periodontal tissues damage. Different strategies have been explored for reinforcing the therapeutic effect of non-surgical periodontal treatment of periodontal tissue damage.
The reviewer is missing in the publication an intensive literature discussion due to the topic “Different strategies have been explored for reinforcing the therapeutic effect of non-surgical periodontal treatment of periodontal tissue damage”. a short pubmed research would be very helpful here. So, any reference to recently published papers on the topic is missing:
- Evaluating clinical and laboratory effects of ozone in non-surgical periodontal treatment: a randomized controlled trial. Seydanur Dengizek E, Serkan D, Abubekir E, Aysun Bay K, Onder O, Arife C.J Appl Oral Sci. 2019 Jan 14;27: e20180108. doi: 10.1590/1678-7757-2018-0108.
- The effects of ozone therapy as an adjunct to the surgical treatment of peri-implantitis. Isler SC, Unsal B, Soysal F, Ozcan G, Peker E, Karaca IR. J Periodontal Implant Sci. 2018 Jun 30;48(3):136-151. doi: 10.5051/jpis.2018.48.3.136. eCollection 2018 Jun. PMID: 29984044
- Thank you so much for your considerations. About the discussion of “Different strategies have been explored for reinforcing the therapeutic effect of non-surgical periodontal treatment of periodontal tissue damage”, we referred to e.g. laser therapy or an adjunct of antibacterial agents etc. Then, we wanted only to reinforce the concept that different strategies have been explored to support the periodontal therapy, but we didn’t aim to examine the differences between these options. We think that the intensive discussion about these concepts may be misleading respect our aim. The references have been added and discussed, as suggested, as limitation of our study:
“…Although significant differences in CAL between the groups, some factors could be considered, plausible interindividual differences and the time between ozone treatment and the subsequent 3 months could reveal a different healing pattern in these individuals. The addition of ozone treatment showed a marked improvement in periodontal conditions compared with the test group, while both groups manifested a significant reduction in pocket probing at 3 months. Each patient received the same oral hygiene instruction and motivation, and this may influence long-term outcomes. The current study has some limitations, first, the use of ozone therapy was only granted during the SRP phase, without any recall, and the follow-up was limited to explore the potential benefit of recall. The limitation of this study could be related to the use of criteria to define success as changes in PD and CAL, because of the potential limited representativeness of the effectiveness of ozone therapy, which includes additional benefits. Further studies investigating biochemical parameters of oxidative stress might be useful for a more in-depth evaluation on periodontal tissue healing.”
- To test this hypothesis, the authors conducted a prospective randomized study comparing non-surgical perio-dontal treatment plus gaseous ozone therapy to standard therapy. A total of 90 healthy individuals with moderate or severe generalized periodontitis were involved in the study. The trial was conducted from September 2019 to October 2020. Forty-five patients were randomized to receive scaling and root-planning (SRP) used as conventional non-surgical periodontal therapy plus gaseous ozone therapy (GROUP A); forty-five were allocated to standard treatment (GROUP B). The endpoint was defined as the periodontal response rate after the application of the ozone therapy at 3 months and 6 months, defined as no longer meeting the criteria for active periodontitis. Statistical analysis was performed employing SPSS v.18 Chicago: SPSS Inc. Comparing with the literature the reviewer is missing to evaluate both, clinical and biochemical in vivo effects of gaseous ozone on oxidative stress (TAS and TOS) and TGF-β, a marker for periodontal recovery.
About periodontal healing, the parameters used in all studies are probing pocket depth and clinical attachment level. These are crucial clinical parameters that alone allow me to determine periodontal health. In daily clinical management, the goal of non-surgical periodontal therapy is to reduce the above parameters. The following two articles serve only as an example: Preus HR, Gjermo P, Baelum V. A Randomized Double-Masked Clinical Trial Comparing Four Periodontitis Treatment Strategies: 5-Year Tooth Loss Results. J Periodontol. 2017 Feb;88(2):144-152. doi: 10.1902/jop.2016.160332. Epub 2016 Oct 21. PMID: 27767387; Preus HR, Gunleiksrud TM, Sandvik L, Gjermo P, Baelum V. A randomized, double-masked clinical trial comparing four periodontitis treatment strategies: 1-year clinical results. J Periodontol. 2013 Aug;84(8):1075-86. doi: 10.1902/jop.2012.120400. Epub 2012 Oct 29. PMID: 23106511). Further, the articles that you refer specify their aim: “to evaluate the clinical and biochemical (oxidative stress and pro-inflammatory mediators) effects of the gaseous ozone use accompanied by scaling and root planning (SRP) in periodontal treatment”. Our aim was to evaluate only the clinical parameters. We had a different objective and it cannot a discriminant.
- As results the authors revealed that periodontal parameters differed significantly between patients treated with the two distinct procedures at 3 months (P= <0.005); a statistically significant difference between groups was observed from base-line in the CAL (P= <0.0001), PPD (P= <0.0001) and BOP (P= <0.0001) scores. The reviewer agrees the use of ozone in periodontal therapy is based on its antimicrobial, immunostimulating, anti-hypoxic, and biosynthetic properties. Ozone is theorized to contribute to periodontal healing by eliminating pathogens, activating the immune system, and stimulating the humoral antioxidant system, however, clinical evidence thereto is limited. The reviewer is missing a clear outline of the clinical trials strategy in this paper. The reviewer is convinced that the study has been done using a randomized, double-blinded study design.
- Potentially, more parameters could be evaluated than those examined, but our goal was not to evaluate oxidative stress, but clinical parameters as we have highlighted. The sentence concerning the effects of tissue ozone has been reported because we inevitably must explain what ozone therapy is for and what its function is in terms of effectiveness. However, this does not mean that our goal is to demonstrate the effects on oxidative stress. Our goal was not to demonstrate the effects on oxidative stress, but in general on tissue healing compared to conventional treatment only on a clinical basis. We did not set the study on biochemical variables. Mentioning the capabilities of ozone does not necessarily imply examination of biochemical parameters. We think it is an excellent cue for future analysis but does not imply inferiority of the completed study. The outcomes evaluated were others. Let us give you an example: multiple similar clinical studies concerning, for example, the use of the laser, explain that the laser has effects on fibroblasts and epithelium. However, none of these studies investigated other parameters (e.g. histological evaluation) beyond periodontal parameters. Yet histologic examination could have confirmed what was claimed. They are firm to periodontal clinical parameters.
- The study design has been implemented with a more complete description, as follows:
- In conclusions the present study suggests that SRP combined with ozone therapy in the treatment of periodontitis revealed an improved outcome than SRP alone. Ozone has several positive effects on cellular and humoral immune system components, in stimulation of the proliferation of immunocompetent cells and synthesis of immunoglobulins. It has been well-established that ozone activates macrophages and enhances the sensitivity of microorganisms to phagocytosis. Immune cells of the body produce certain cytokines as a response to the activation caused by ozone. Ozone triggers the synthesis of various biologically active substances that reduce inflammation and promote wound healing, such as interleukins, leukotrienes, and prostaglandins. Oxidant and AO molecules are also known to play important roles in the immune response to PD. For this purpose, the authors should evaluate the effects of ozone treatment on oxidants and AOs by assaying levels of TOS, NO, 8-OHdG, MDA, GSH, and TAS in saliva to investigate the ozone effects on the immune response in PD. These newly investigated findings may suggest that the stimulatory effects of ozone administration on periodontal tissues may be limited.
- Potentially, more parameters could be evaluated than those examined, but our goal was not to evaluate oxidative stress, but clinical parameters as we have highlighted. The sentence concerning the effects of tissue ozone has been reported because we inevitably must explain what ozone therapy is for and what its function is in terms of effectiveness. However, this does not mean that our goal is to demonstrate the effects on oxidative stress. Our goal was not to demonstrate the effects on oxidative stress, but in general on tissue healing compared to conventional treatment only on a clinical basis. We did not set the study on biochemical variables. Mentioning the capabilities of ozone does not necessarily imply examination of biochemical parameters. We think it is an excellent cue for future analysis but does not imply inferiority of the completed study. The outcomes evaluated were others. Let us give you an example: multiple similar clinical studies concerning, for example, the use of the laser, explain that the laser has effects on fibroblasts and epithelium. However, none of these studies investigated other parameters (e.g. histological evaluation) beyond periodontal parameters. Yet histologic examination could have confirmed what was claimed. They are firm to periodontal clinical parameters.
Reviewer 3 Report
Thank you so much for letting me review this manuscript, it needs some revisions.
Correct abstract
Correctly entered keywords
Correct materials and methods and statistical analysis
Discussion, limitations must be added, in order to maintain an optimal periodontal state of health for a correct vision of the periodontal ligament it is necessary to motivate the patient to a correct home management through the use of a roto oscillating or sonic toothbrush and toothpastes using hyaluronic acid for keep the periodontium intact and avoid the progression of the disease with the destruction of the tissue itself. I add reference:
DOI 10.3390/app11188586
DOI 10.3390/ijerph18041468
Correct conclusions
Author Response
#Reviever 3
- Thank you so much for letting me review this manuscript, it needs some revisions.
Correct abstract
Correctly entered keywords
Correct materials and methods and statistical analysis
Discussion, limitations must be added, in order to maintain an optimal periodontal state of health for a correct vision of the periodontal ligament it is necessary to motivate the patient to a correct home management through the use of a roto oscillating or sonic toothbrush and toothpastes using hyaluronic acid for keep the periodontium intact and avoid the progression of the disease with the destruction of the tissue itself. I add reference:
DOI 10.3390/app11188586
DOI 10.3390/ijerph18041468
Thank you so much for your considerations. We motivated our patients to use the roto oscillating or sonic toothbrush. Then, we added this component, as follows: “To maintain a state of optimal periodontal health for a correct view of the periodontal ligament, each patient was motivated to a correct home management using a roto oscillating or sonic toothbrush and toothpastes to keep the periodontium intact and avoid the progression of the disease with the destruction of the tissue itself.” The references have been added.
Correct conclusions
Reviewer 4 Report
The study is of confirmatory interest. Ozone therapy is not new but in my opinion, confirming findings is a good thing.
There are two problems with this manuscript: although some improvements are reported in clinical parameters, the difference in CAL as initially detected in both groups remains. This must be addressed by the authors.
Then, as two sub-groups are included (moderate and severe periodontitis), are results influenced by this fact? Would ozone therapy lead to more improvements in one of the groups presenting moderate or severe periodontitis? This can be evaluated using a more in-depth statistical analysis.
This brings me to the statistics, my hobby-horse:
In the tables an indiscriminate use of statistical parameters (mean, median, SD, 95% CI just looks like a copy and paste from SPSS outcome. Rationale for using all these parameters together must either be given or a choice must be made.
In giving number of decimals, make a choice according to the relevance. Example: if you have 90 patients, 1 patient would roughly be 1%. So giving data as 10.12% is meaningless as there are no 1/100th of a patient. Same applies for PPD or CAL: measured with a standard periodontal probe, at best 0.5mm would be the discriminatory limit of this instrument, so give data rounded accordingly.
In the tables, a p-value is stated but not the type of test. In my opinion, giving a prportion between male/female, sever/moderate periodontits cannot be evaluated using a t-test.
And, as stated above, there are enough participants to at least try a more sophisticated analysis using covariates such as age, gender, severity on the different outcomes.
It should alo be stated in which site the study was performed and the number and qualification of practitioners involved.
Was there an evaluation period between initial treatment and start of SRP with or without ozone? If yes, how long?
Discussion: describe strong and weak points of your study before starting to compare with the literature.
Author Response
#Reviever 4
- The study is of confirmatory interest. Ozone therapy is not new but in my opinion, confirming findings is a good thing. There are two problems with this manuscript: although some improvements are reported in clinical parameters, the difference in CAL as initially detected in both groups remains. This must be addressed by the authors.
Thank you so much for your considerations.
Improvement in clinical parameters was shown in both groups. The difference in CAL, based on our results and study design, is due precisely to the treatment. CAL is a periodontal parameter that together with PPD allows us to monitor improvements and this is evidenced in the results. The difference in CAL could be explained by the possibility that treatment with ozone favoured the establishment of greater epithelial attachment in the adjuvant-treated group and was desirable.
- Then, as two sub-groups are included (moderate and severe periodontitis), are results influenced by this fact? Would ozone therapy lead to more improvements in one of the groups presenting moderate or severe periodontitis? This can be evaluated using a more in-depth statistical analysis.
Thank you so much for your considerations.
Statistical analysis which included the impact of the stage of disease has been added, as required in the further observation (point 7).
- This brings me to the statistics, my hobby-horse: In the tables an indiscriminate use of statistical parameters (mean, median, SD, 95% CI just looks like a copy and paste from SPSS outcome. Rationale for using all these parameters together must either be given or a choice must be made.
- Thank you for your considerations. The correction has been applied, by removing of median. The choice was described, specifying that we conducted a descriptive analysis and the Unpaired t Test to compare the results over each time point.
- In giving number of decimals, make a choice according to the relevance. Example: if you have 90 patients, 1 patient would roughly be 1%. So giving data as 10.12% is meaningless as there are no 1/100th of a patient. Same applies for PPD or CAL: measured with a standard periodontal probe, at best 0.5mm would be the discriminatory limit of this instrument, so give data rounded accordingly.
Thank you so much for your considerations.
- For PPD and CAL values, we chose not to round as in most periodontal studies. Here are some excerpts from some articles of the journals of periodontology in which the values of CAL and PPD are not rounded:
Preus HR, Gunleiksrud TM, Sandvik L, Gjermo P, Baelum V. A randomized, double-masked clinical trial comparing four periodontitis treatment strategies: 1-year clinical results. J Periodontol. 2013 Aug;84(8):1075-86. doi: 10.1902/jop.2012.120400. Epub 2012 Oct 29. PMID: 23106511.
Preus HR, Gjermo P, Baelum V. A Randomized Double-Masked Clinical Trial Comparing Four Periodontitis Treatment Strategies: 5-Year Tooth Loss Results. J Periodontol. 2017 Feb;88(2):144-152. doi: 10.1902/jop.2016.160332. Epub 2016 Oct 21. PMID: 27767387.
- In my opinion, giving a prportion between male/female, sever/moderate periodontits cannot be evaluated using a t-test:
Thank you so much for your considerations.
The correction has been applied and the column has been deleted.
- In the tables, a p-value is stated but not the type of test
Thank you so much for your considerations.
-I’m sorry, but we described that the Unpaired t Test was used to compare the results over each time point
- It should also be stated in which site the study was performed and the number and qualification of practitioners involved.
Thank you so much for your considerations.
The requested informations have been added, as follows: “Periodontal examinations were performed by one blinded examiner (BR), and three operators (dentist, AS; dentist, FI, dental hygienist, EF) carried out the treatment at each time point. One blinded statistician (AN) performed the data analysis”
- The site of study and the number and qualification of practitioners have been added, as follows:
- Was there an evaluation period between initial treatment and start of SRP with or without ozone? If yes, how long?
Thank you so much for your considerations.
- Patients were recalled, to record periodontal clinical parameters at 3 and 6 months, as described. The initial treatment is coincident with the start of SRP with or without ozone. No formal interim analysis or interim statistical testing for treatment comparisons was planned. Examinations ad interim are not possible, because of biological reasons. Control probing is not allowed because it would risk damaging periodontal tissues and fibres that are in the early stages of healing.
- And, as stated above, there are enough participants to at least try a more sophisticated analysis using covariates such as age, gender, severity on the different outcomes.
Thank you so much for your considerations.
- As suggested, we have performed an additional statistical analysis, as follows:
Statistical analysis
A mixed model multivariate analysis of covariance (MANCOVA) with two within-subjects factors and one between-subjects factor was conducted to determine whether significant differences exist among the time points for PPD and CAL between the levels of treatment (SRP+ OZONE or SRP) after controlling for stage of disease (moderate or severe), age, and sex as covariates. A p value of <0.05 was considered statistically significant. Continuous variables were presented as mean and standard deviation (SD). The comparison of data from the two groups at each time point was performed by using the unpaired 2-sample t test.
3.1.1 Assumptions
Normality. The assumption of normality was assessed by plotting the quantiles of the model residuals against the quantiles of a Chi-square distribution. Figure 3 presents a Q-Q scatterplot of model residuals.
Figure 3. Q-Q scatterplot for normality of the residuals for the regression model.
Homoscedasticity. Homoscedasticity was evaluated by plotting the residuals against the predicted values (Bates et al., 2014; Field, 2017; Osborne & Walters, 2002). Figure 4 presents a scatterplot of predicted values and model residuals.
Figure 4. Residuals scatterplot testing homoscedasticity.
Sphericity. Mauchly's test was used to assess the assumption of sphericity (Field, 2017; Mauchly, 1940). The results showed that the variances of difference scores across the levels of Time Factor were all similar based on an alpha of 0.05, p = .483, indicating the sphericity assumption was met for Time Factor. The results showed that the variances of difference scores across the levels of Time Factor:Dipendent variable (Dv) Factor were significantly different from one another based on an alpha of 0.05, p < .001, indicating the sphericity assumption was violated for Time Factor:Dv Factor.
Multivariate Outliers. To identify influential points in the residuals, Mahalanobis distances were calculated and compared to a χ2 distribution (Newton & Rudestam, 2012). An outlier was defined as any Mahalanobis distance that exceeds 22.46, the 0.999 quantile of a χ2 distribution with 6 degrees of freedom (Kline, 2015). There were no outliers detected in the model.
Homogeneity of regression slopes. The assumption for homogeneity of regression slopes was assessed by rerunning the mixed model MANCOVA, but this time including interaction terms between each independent variable and covariate (Field, 2017; Pituch & Stevens, 2015). The model with covariate-independent variable interactions did not explain significantly more variance in the dependent variables than the original model, F(18, 207) = 1.1, p = .356. This implies that none of the covariates interacted with the independent variables and the assumption of homogeneity of regression slopes was met.
Covariate-IV independence. An ANOVA was conducted for each pair of numeric covariates and independent variables to assess independence (Field, 2017). A multinomial regression model was conducted and compared to the null model for each pair of categorical covariates and independent variables to assess independence. There were no significant models for any combination of covariates and independent variables based on an alpha of 0.05, indicating the assumption of independence between covariates and independent variables was met.
Results
The results were examined based on an alpha of 0.05. Table 6 presents the MANCOVA results.
Table 6. Mixed Model MANCOVA Results.
|
Source |
df |
SS |
MS |
F |
p |
ηp2 |
|
Between-Subjects |
|
|
|
|
|
|
|
Treatment |
1 |
8.76 |
8.76 |
23.28 |
< .001 |
0.24 |
|
Stage_of_disease |
1 |
1.78 |
1.78 |
4.73 |
.033 |
0.06 |
|
Age |
1 |
2.90 |
2.90 |
7.69 |
.007 |
0.09 |
|
sex |
1 |
0.92 |
0.92 |
2.43 |
.123 |
0.03 |
|
Residuals |
75 |
28.24 |
0.38 |
|
|
|
|
|
|
|
|
|
|
|
|
Within-Subjects |
|
|
|
|
|
|
|
Time Factor |
2 |
4.41 |
2.20 |
8.11 |
< .001 |
0.10 |
|
Treatment:Time Factor |
2 |
13.82 |
6.91 |
25.45 |
< .001 |
0.25 |
|
Stage_of_disease:Time Factor |
2 |
0.99 |
0.49 |
1.82 |
.166 |
0.02 |
|
Age:Time Factor |
2 |
0.40 |
0.20 |
0.73 |
.482 |
0.01 |
|
sex:Time Factor |
2 |
0.71 |
0.36 |
1.32 |
.271 |
0.02 |
|
Time Factor Residuals |
150 |
40.74 |
0.27 |
|
|
|
|
|
|
|
|
|
|
|
|
Dv Factor |
1 |
0.23 |
0.23 |
0.86 |
.358 |
0.01 |
|
Treatment:Dv Factor |
1 |
2.38 |
2.38 |
8.74 |
.004 |
0.10 |
|
Stage_of_disease:Dv Factor |
1 |
0.07 |
0.07 |
0.26 |
.611 |
0.00 |
|
Age:Dv Factor |
1 |
0.04 |
0.04 |
0.15 |
.698 |
0.00 |
|
sex:Dv Factor |
1 |
0.13 |
0.13 |
0.47 |
.494 |
0.01 |
|
Dv Factor Residuals |
75 |
20.43 |
0.27 |
|
|
|
|
|
|
|
|
|
|
|
|
Time Factor:Dv Factor |
2 |
0.11 |
0.06 |
0.20 |
.778 |
0.00 |
|
Treatment:Time Factor:Dv Factor |
2 |
5.43 |
2.71 |
9.57 |
< .001 |
0.11 |
|
Stage_of_disease:Time Factor:Dv Factor |
2 |
0.01 |
0.01 |
0.03 |
.955 |
0.00 |
|
Age:Time Factor:Dv Factor |
2 |
0.24 |
0.12 |
0.42 |
.617 |
0.01 |
|
sex:Time Factor:Dv Factor |
2 |
0.73 |
0.37 |
1.29 |
.276 |
0.02 |
|
Time Factor:Dv Factor Residuals |
150 |
42.56 |
0.28 |
|
|
|
Degrees of Freedom (df): Refers to the number of values used to compute a statistic; an F-test has two values for df: the first is determined by the number of groups being compared - 1, and the second is approximately the number of observations in the sample; used with the F to determine the p-value; F Ratio (F): The ratio of explained variance to error variance; used with the two df values to determine the p-value; Partial Eta Squared (η2p): Effect size for the ANOVA/MANOVA and determines the strength of the differences among the groups; p-value: The probability of obtaining the observed results if the null hypothesis is true; Residuals:Refers to the difference between the predicted value for the dependent variable and the actual value of the dependent variable.
The p-values for and any interaction with these within-subjects factors were calculated using the Greenhouse-Geisser corrections to adjust for the violation of the sphericity assumption.
Between-Subjects. The main effect for Treatment was significant F(1, 75) = 23.28, p < .001, indicating that there were significant differences in PPD and CAL between the levels of Treatment after controlling for stage of disease, age, and sex. The covariate, Stage_of_disease, was significantly related to PPD and CAL, F(1, 75) = 4.73, p = .033. The covariate, age, was significantly related to PPD and CAL, F(1, 75) = 7.69, p = .007. The covariate, sex, was not significantly related to PPD and CAL, F(1, 75) = 2.43, p = .123.
Within-Subjects. The main effect for Time Factor was significant F(2, 150) = 8.11, p < .001, indicating there were significant differences in PPD and CAL across the levels of Time Factor ignoring Dv Factor after controlling for stage of disease, age, and sex. The main effect for Dv Factor was not significant F(1, 75) = 0.86, p = .358, indicating the values for across the levels of Dv Factor, PPD and CAL, were all similar regardless of Time Factor after controlling for stage of disease, age, and sex. The main effect for Time Factor and Dv Factor was not significant F(2, 150) = 0.20, p = .778, indicating that the relationships between the levels of Dv Factor were similar across the levels of Time Factor after controlling for Stage of disease, age, and sex.
Within-Between Interactions. The interaction effect between Time Factor and treatment was significant F(2, 150) = 25.45, p < .001, indicating that the relationships between the levels of Time Factor differed significantly between the levels of treatment ignoring Dv Factor after controlling for stage of disease, age, and sex.
The interaction effect between Dv Factor and Treatment was significant F(1, 75) = 8.74, p = .004, indicating that the relationships between the levels of Dv Factor differed significantly between the levels of treatment regardless of Time Factor after controlling for stage of disease, Age, and sex.
The interaction effect between Time Factor, Dv Factor, and treatment was significant F(2, 150) = 9.57, p < .001, indicating that the relationships between the combinations of Time Factor and Dv Factor differed significantly between the levels of treatment after controlling for stage of disease, age, and sex.
Within-Covariate Interactions. The interaction effect between Time Factor and stage of disease was not significant, F(2, 150) = 1.82, p = .166, indicating that the relationships between the levels of Time Factor were similar for all values of stage of disease. The interaction effect between Time Factor and age was not significant, F(2, 150) = 0.73, p = .482, indicating that the relationships between the levels of Time Factor were similar for all values of age. The interaction effect between Time Factor and sex was not significant, F(2, 150) = 1.32, p = .271, indicating that the relationships between the levels of Time Factor were similar between the levels of sex.
The interaction effect between Dv Factor and stage of disease was not significant, F(1, 75) = 0.26, p = .611, indicating that the relationships between the levels of Dv Factor were similar for all values of stage of disease. The interaction effect between Dv Factor and age was not significant, F(1, 75) = 0.15, p = .698, indicating that the relationships between the levels of Dv Factor were similar for all values of age. The interaction effect between Dv Factor and sex was not significant, F(1, 75) = 0.47, p = .494, indicating that the relationships between the levels of Dv Factor were similar between the levels of sex.
The interaction effect between Time Factor, Dv Factor, and Stage_of_disease was not significant, F(2, 150) = 0.03, p = .955, indicating that the relationships between the combinations of Time Factor and Dv Factor were similar for all values of stage of disease. The interaction effect between Time Factor, Dv Factor, and age was not significant, F(2, 150) = 0.42, p = .617, indicating that the relationships between the combinations of Time Factor and Dv Factor were similar for all values of age. The interaction effect between Time Factor, Dv Factor, and sex was not significant, F(2, 150) = 1.29, p = .276, indicating that the relationships between the combinations of Time Factor and Dv Factor were similar between the levels of sex.
References
Bates, D., Mächler, M., Bolker, B., & Walker, S. (2014). Fitting linear mixed-effects models using lme4: arXiv preprint arXiv, Journal of Statistical Software. https://doi.org/10.18637/jss.v067.io1
DeCarlo, L. T. (1997). On the meaning and use of kurtosis. Psychological Methods, 2(3), 292-307. https://doi.org/10.1037/1082-989X.2.3.292
Field, A. (2017). Discovering statistics using IBM SPSS statistics: North American edition. Sage Publications
Greenhouse, S. W., & Geisser, S. (1959). On methods in the analysis of profile data. Psychometrika, 24(2), 95-112. https://doi.org/10.1007/BF02289823
Kline, R. B. (2015). Principles and practice of structural equation modeling. Guilford Publications.
Mauchly, J. W. (1940). Significance test for sphericity of a normal n-variate distribution. The Annals of Mathematical Statistics, 11(2), 204-209. https://doi.org/10.1214/aoms/1177731915
Miller, G. A., & Chapman, J. P. (2001). Misunderstanding analysis of covariance. Journal of Abnormal Psychology, 110(1), 40. https://doi.org/10.1037//0021-843x.110.1.40
Osborne, J., & Waters, E. (2002). Four assumptions of multiple regression that researchers should always test. Practical Assessment, Research & Evaluation, 8(2), 1-9.
- Discussion: describe strong and weak points of your study before starting to compare with the literature.
Although significant differences in CAL between the groups, some factors could be considered, plausible interindividual differences and the time between ozone treatment and the subsequent 3 months could reveal a different healing pattern in these individuals. The addition of ozone treatment showed a marked improvement in periodontal conditions compared with the test group, while both groups manifested a significant reduction in pocket probing at 3 months. Each patient received the same oral hygiene instruction and motivation, and this may influence long-term outcomes. The current study has some limitations, first, the use of ozone therapy was only granted during the SRP phase, without any recall, and the follow-up was limited to explore the potential benefit of recall. The limitation of this study could be related to the use of criteria to define success as changes in PD and CAL, because of the potential limited representativeness of the effectiveness of ozone therapy, which includes additional benefits. Further studies investigating biochemical parameters of oxidative stress might be useful for a more in-depth evaluation on periodontal tissue healing.
